# Faint Galaxy Number Counts in the Durham and SDSS Catalogues

John H. Marr

Unit of Computational Science, Babraham Research Campus, Cambridge CB22 3AT, UK; john.marr@2from.com

**Abstract:** Galaxy number counts in the *K*-, *H*-, *I*-, *R*-, *B*- and *U*-bands from the Durham Extragalactic Astronomy and Cosmology catalogue could be well-fitted over their whole range using luminosity function (LF) parameters derived from the SDSS at the bright region and required only modest luminosity evolution with the steepening of the LF slope ($\alpha$), except for a sudden steep increase in the *B*-band and a less steep increase in the *U*-band at faint magnitudes that required a starburst evolutionary model to account for the excess faint number counts. A cosmological model treating Hubble expansion as an Einstein curvature required less correction at faint magnitudes than a standard ΛCDM model, without requiring dark matter or dark energy. Data from DR17 of the SDSS in the *g*, *i*, *r*, *u* and *z* bands over two areas of the sky centred on the North Galactic Cap (NGC) and above the South Galactic Cap (SGC), with areas of 5954 and 859 sq. deg., respectively, and a combined count of 622,121 galaxies, were used to construct bright galaxy number counts and galaxy redshift/density plots within the limits of redshift $\leq 0.4$ and mag $\leq 20$. Their comparative densities confirmed an extensive void in the Southern sky with a deficit of 26% out to a redshift $z \leq 0.15$. Although not included in the number count data set because of its incompleteness at fainter magnitudes, extending the SDSS redshift-number count survey to fainter and more distant galaxies with redshift $\leq 1.20$ showed a secondary peak in the number counts with many QSOs, bright X-ray and radio sources, and evolving irregular galaxies with rapid star formation rates. This sub-population at redshifts of 0.45–0.65 may account for the excess counts observed in the *B*-band. Recent observations from the HST and James Webb Space Telescope (JWST) have also begun to reveal a high density of massive galaxies at high redshifts ($z > 7$) with high UV and X-ray emissions, and future observations by the JWST may reveal the assembly of galaxies in the early universe going back to the first light in the universe.

**Keywords:** galaxies: number counts; galaxies: evolution; general relativity

## 1. Introduction

Galaxy number counts (GNCs) as a function of magnitude provided an early, straightforward quantitative measurement in cosmology, and numerous surveys have continued to increase the faint magnitude limit in six principal observational photometric bands (K, H, I, R, B and U). The faint magnitude depth has been regularly extended over the years for each spectral band, and many individual series of observations have been collated and published by the Extragalactic Astronomy and Cosmology Research Group of Durham University [1]. These observational data for GNCs have been accumulated over many years from a variety of sources and across different sky fields, causing much of the data to be widespread, and GNCs in the six principal optical bands have traditionally been plotted as scatter plots to include each individual data point. One large source of error is that each bin contains galaxies with a range of morphologies and redshifts and, hence, a range of ages and evolutionary histories, so individual bins may require a range of corrections that have to be averaged for each bin, but attempts to clarify these averaged counts by quantifying them by morphology and redshift defeated the essential simplicity of complete magnitude counts. All counts were binned into 0.5 magnitude bins, with error bars to indicate the range of observations included in each bin, resulting in cleaner data points for each band.

The different rates of evolution in past epochs, which appear to have gradually increased with decreasing observational wavelength from the *K*- to *U*-bands and the excess numbers in the *B*-band and, to a lesser extent in the *U*-band, were clearly evident. Metcalfe et al. [2] suggested that this may be from a second peak at high redshift, possibly explained by a sub-population of early-type galaxies with ongoing star formation. Arnouts et al. [3] presented measurements of the galaxy LF at 1500 Å in the range $0.2 \leq z \leq 1.2$ based on Galaxy Evolution Explorer VIMOS-VLT Deep Survey observations for 1000 spectroscopic redshifts and at higher $z$ using existing data sets. Their main results were: (a) luminosity evolution is observed with $\Delta M^* \sim -2.0$ mag in the range $0 \leq z \leq 1$ and $\Delta M^* \sim -1.0$ mag in the range $1 \leq z \leq 3$, confirming that star formation activity was significantly higher in the past; (b) the LF slopes vary in the range $-1.2 \geq \alpha \geq -1.65$, with a possible further increase at higher $z$; and (c) the analysis of three spectral-type classifications, Sb-Sd, SdIrr and unobscured starbursts found that, although the bluest class evolved less strongly in luminosity than the other two classes, their number density increased sharply with $z$ from $\simeq$15% in the local universe to $\simeq$55% at $z \simeq 1$, while that of the reddest classes decreased.

Since the launch of the Hubble Space Telescope in 1990, there has been a wealth of fresh data from newer deep-space telescopes and the JWST, which has begun to report observations of galaxies in the far-infrared at deep magnitudes. The Hubble Space Telescope Medium Deep Survey (HST-MDS) found a dwarf-rich population at $z = 0.3$–0.5 [4], and a number of recent surveys have identified galaxies with stellar masses as high as $\sim 10^{11} M_\odot$ out to redshifts $z \simeq 10$. This paper aims to consider how some of the newly published deep redshift data fit into the interpretation of the faint GNCs.

## 2. Calculating the Observed Volume

Number counts are determined in increments of apparent magnitude or half-magnitude per square degree of observed sky; however, space is curved through the expansion of the universe, and the intrinsic luminosity and density of galaxies are not uniform [5]. Because magnitude is a function of the luminosity distance ($D_L$), whereas the observable area is a function of the diameter distance ($D_A$), the volume increment must be parameterised in $z$, producing a wineglass-shaped observable volume (Figure 1). The volume element/sterad is:

$$\delta V = D_H^3 D_A^2 \delta D_L, \tag{1}$$

where $D_H$ is the Hubble distance, $D_A$ is the angular diameter distance and $D_L$ is the luminosity distance, and

$$\begin{aligned} D_H &= c/H_0 \\ D_A &= D_C/(1+z) \\ D_L &= D_C(1+z). \end{aligned} \tag{2}$$

$D_C$ is the line-of-sight comoving distance, generally obtained by integrating the comoving equation for a Λ-Cold Dark Matter (ΛCDM) line-of-sight model derived from general relativity (GR) and involving terms for dark matter and dark energy. The model for this paper included a curvature term for Hubble expansion to incorporate the curvature of light across an expanding universe [6]. This has been shown to closely approximate a standard ΛCDM model for angular diameter distances derived from baryonic acoustic oscillation (BAO) data [7] and for luminosity distances derived from extensive supernova (SNe 1a) data [8]. By letting $\Omega_\Lambda = 0$ and assuming the intrinsic curvature term $\Omega_K = 0$, this has an analytical solution in $\Omega_m$ and $z$ (Equation (3)):

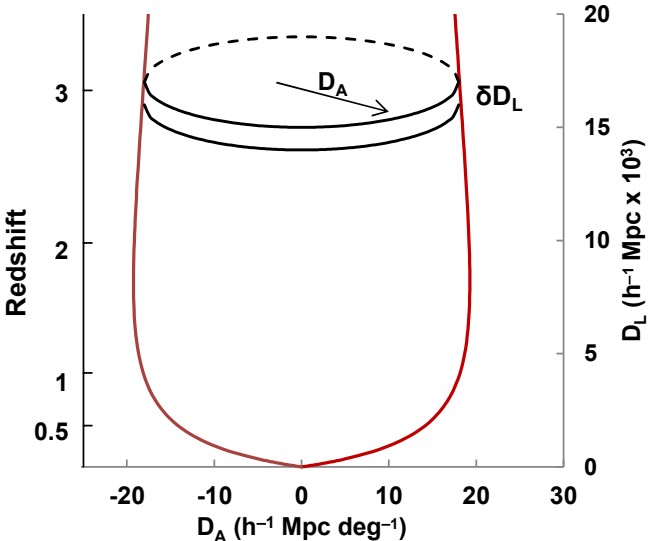

**Figure 1.** A graphic showing the observed volume/square degree with redshift.

$$D_C = \frac{1}{\sqrt{1 - \Omega_m}} \log \left( \frac{(1 + z)\left((1 - 0.5\Omega_m) + \sqrt{1 - \Omega_m}\,\right)}{1 + 0.5\Omega_m(z - 1) + \sqrt{(1 - \Omega_m)(1 + \Omega_m z)}} \right) \tag{3}$$

$\delta V$ may now be expressed for the GR model in terms of $z$ as:

$$\delta V = D_H^3 \frac{D_C^2}{(1 + z)^2} \frac{d}{dz}[D_C(1 + z)]\delta z \quad [h^3 \text{ Mpc}^3 \text{ sterad}^{-1}], \tag{4}$$

while the density function must be multiplied by a further factor of $(1 + z)^3$ to allow for the contraction of space in three dimensions with look-back time. Integrating to $z_{max}$, the redshift limit for the integral, gives the total volume per square degree. The number of galaxies per square degree per magnitude increment is then:

$$n(m) = D_H^3 h^{-3}(1/3283.8) \int_0^{z_{max}} \Phi_z(m)\delta V \, dz \, dm. \tag{5}$$

where $\Phi_z(m)$ is the luminosity function (Section 2.1). $z_{max}$ is taken to be the volume limit for the integral, but, like much else in magnitude-number count analysis, this limit is debated. Results from the Wilkinson Microwave Anisotropy Probe (WMAP) suggest reionisation at $z$ = 11–30 [9], but an emission signature of broad and asymmetric Lyman-$\alpha$ emission may specify the redshift of the final stages of reionisation, suggesting a redshift range $z$ = 6–9 [10]; however, counts of bright galaxies at redshifts $z \sim 6$ by Lyman-break selection find too few luminous galaxies for their hot stellar populations to power reionisation [11]. This makes the most likely culprit lower-luminosity galaxies, which may equate to the small, low-metallicity objects found in significant numbers at $z \sim 2$–3 and sometimes termed subgalactic [12].

### 2.1. The Schechter Luminosity Function (LF)

For a specific sample of galaxies, $S$, it is usual to refer to the luminosity distribution defined by Schechter [13] as $ns(L)$ galaxies per unit luminosity for the sample. For a given sample size, the volume of the sample will vary with the luminosity and can be defined as $V_s(L)$.

Then, the number of galaxies in $S$ in the luminosity interval $\delta L$ centred on $L$ is $n_S(L)\delta L$, and the luminosity function (LF) $\Phi_S(L)$ of the sample $S$ has the units of number of galaxies

per unit luminosity per unit volume and is defined as $\Phi_S(L) \equiv \Phi(L)\delta L \, V_S(L)$, which is reached in the limit of $V_S(L) \to \infty$. Schechter defined his luminosity function as

$$\Phi(L)\,\mathrm{d}L = \Phi^*(L/L^*)^\alpha \exp\left(-L/L^*\right) \mathrm{d}(L/L^*) \tag{6}$$

but it is more useful to rewrite Equation (6) in terms of absolute magnitude $M^*$:

$$n_S(M)\,\mathrm{d}M = \Phi^* V_C K \, \mathrm{dex}[0.4(\alpha+1)(M^*-M)] \exp[-\,\mathrm{dex}[0.4(M^*-M)]]\,\mathrm{d}M \tag{7}$$

Although there is considerable variation in the actual parameters, the Schechter function has received good experimental confirmation, and there has also been some theoretical justification from a model of the non-linear matter distribution of rich galaxy clusters and their correlations, showing that the luminosity function has the Schechter form [14]. Nevertheless, the actual form of the Schechter function is an amalgam of galaxies of many types, ages and distances and prone to wide variation, and it has been continuously refined for several band-passes and galaxy types. Folkes et al. defined the function in terms of the spectral type from the Anglo-Australian Observatory 2dFGRS (Two-degree-Field Galaxy Redshift Survey) [15], finding a range of values for the three parameters, $M^*$, $\phi$, and $\alpha$, and Metcalfe et al. described Schechter parameters for five different galaxy types (E/S0, Sab, Sbc, Scd and Sdm) to produce a composite function [2].

### 2.2. Deriving the Number Count Curves

The absolute magnitude $M$ to produce each band of apparent magnitude $m$ over the range of $D_L$ is derived from the definition of $D_L$ in the chosen GR model:

$$M = m - 5\log_{10}(D_L) - 25 \tag{8}$$

where $D_L = D_H(1+z)D_C$ for the appropriate GR model. $n(m)$ is then derived by substituting for $M$ in $\phi(z)$ in Equation (5) and integrating Equation (5) to $z_{max}$ and over the range $m \pm 0.5$ per magnitude interval (or $m \pm 0.25$ per half-magnitude interval). The resultant curve is an asymmetric hyperbola (Figure 2) with two distinct asymptotic slopes: the classical $\mathrm{d}\log_{10} N/\,\mathrm{d}m = 0.6$ Euclidean slope at bright magnitudes and a slope of $-0.4(\alpha+1)$ at faint magnitudes. Normalisation over the Euclidean region is determined by $\phi^*$ and $M^*$. The asymmetric point of inflexion is a function of $M^*$ and $\log(z_{max})$ that may be determined by numerical computation, but approximates to:

$$m_{inflexion} \simeq M^* + 2.5\log(z_{max}) + \text{constant}\,. \tag{9}$$

These contributions make no allowance for evolution. In practice, they will show some change with redshift due to evolution from events such as starbursts, luminosity evolution and mergers, making the true underlying cause for any change in slope difficult to identify. Additionally, the observed counts may tail off at the faintest magnitudes, which would depress the faint end of the resultant curve. Blanton's curves in Figure 3 are essentially flat at all bands at the faint end, corresponding to $\alpha \simeq -1.0$. Such flat LF curves give rise to very shallow number count curves that do not reflect reality; to account for the observations, there must be evolution of some or all components of the LF with increasing redshift, as suggested in Figure 2, which combines redshift bands to form the single composite curve with $\alpha = -1.5$.

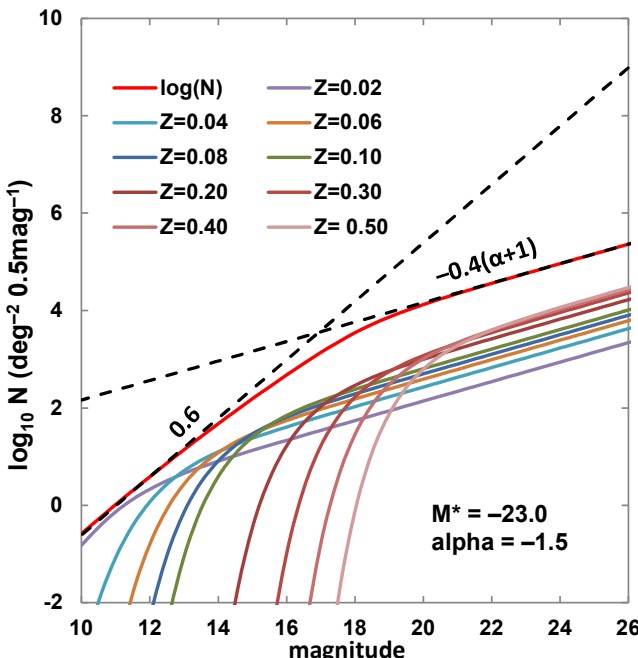

**Figure 2.** Building a theoretical number count curve (red line) with composite Schechter curves at increasing redshifts ($M^* = -23.0; \alpha = -1.5$). The transition from a Euclidean slope of 0.6 to the $\alpha$-dependent slope is clearly shown (dashed lines).

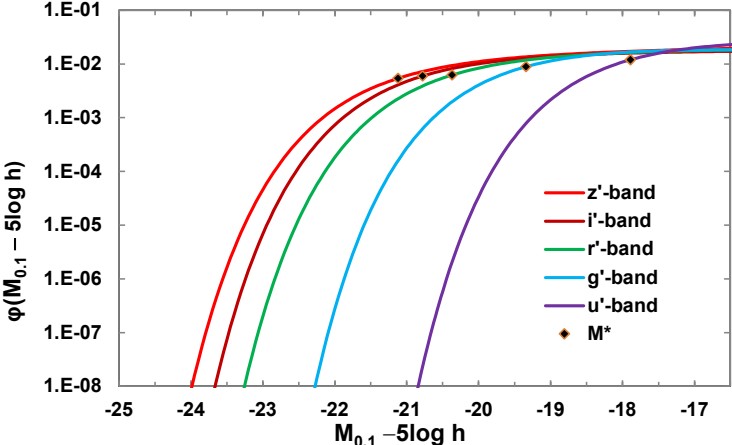

**Figure 3.** Schechter curves for 5 bands at uniform redshift ($z = 0.11$), after Blanton et al. [16].

## 3. Observational Data

Since its first data release in 2003 until the most recent data release of DR17, which became publicly available in December 2021 [17], the Sloan Digital Sky Survey (SDSS) has provided increasing quantities of data with detailed information on a large number of galaxies across five colour bands ($u$, $g$, $r$, $i$ and $z$) out to apparent magnitude limits of ~21–23. This has produced clean, well-defined slopes for the bright galaxy counts. It should be noted that the uppercase bands of the optical bands differ in their spectral ranges from the lowercase designations of the SDSS bands, and their correlation with earlier optical data used a standard conversion to the older $B_J$-band to tie the deeper faint data to these more recent bright galaxy magnitudes.

### 3.1. Observations from the SDSS

The Sloan Digital Sky Survey (SDSS) has provided a wealth of data over five spectral bands, with spectra and usable redshifts for ~2.6 million unique galaxies out to limiting magnitudes of $u' = 22.3$, $g' = 23.3$, $r' = 23.1$, $i' = 22.3$ and $z' = 20.8$ [17]. Two principal areas are covered by the surveys: (a) the area centred on the North Galactic Centre (NGC) and (b) an area above the South Galactic Centre (SGC) (Table 1, J2000 equatorial coordinates). The counts for the smaller SGC area suggest that it has a lower density of galaxies compared with the NGC area, with a deficit of 26% out to a redshift $z \sim 0.15$ (Figure 4), but with an increased SGC density beyond $z \sim 0.1625$ consistent with a local fluctuation of $2\sigma$ in a large-scale structure. A model for an extensive void was developed by Busswell et al. [18], who noted from the 2dFGRS that the Southern counts with $B < 17$ mag were down by ~30% out to $z = 0.1$ relative to the Northern counts, and this appeared to be relatively homogeneous over its whole range, suggesting it could be modelled by varying the density normalisation $\Phi^*$.

**Table 1.** Parameters for the SDSS number count surveys (redshift $\leq 0.4$, $B_J \leq 20$). J2000 equatorial coordinates.

| Region | RA (h) | Dec (deg) | Area (deg$^2$) | Total Count | Density (gals deg$^{-2}$) |
|--------|--------|-----------|----------------|-------------|---------------------------|
| NGC | 8:00–16:00 | 0–60 | 5954 | 562,196 | 94.42 |
| SGC | −2:00–+2:00 | 0–30 | 859 | 59,925 | 69.76 |

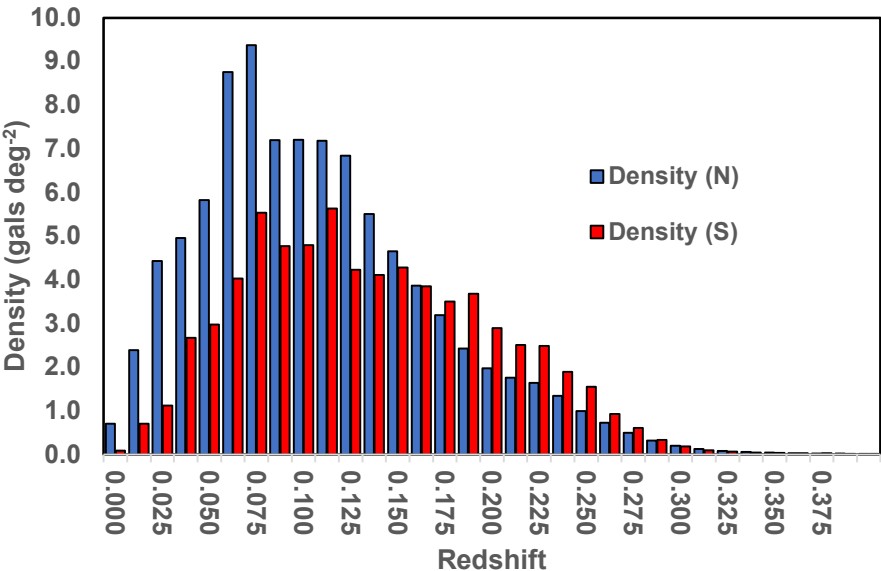

**Figure 4.** SDSS galaxy density in redshift bins of 0.0125. NGC (red bars) and SGC (blue bars). $B_J \leq 20$.

The number density from the SDSS, however, suggests that, beyond $z = 0.15$, the over-density in the Northern counts is reversed to become an under-density compared with the Southern region. In using number count observations, it is therefore important to consider averaging the data from several regions of the sky to compensate for these variations in count density, as presented in Section 4.

### 3.2. The LF from the SDSS

The work of Schechter and others to determine a luminosity function for galaxies distributed randomly through space has been continuously refined. Blanton et al. used data from the extensive SDSS data to define Schechter curves for five observational bands ($^{0.1}u$, $^{0.1}g$, $^{0.1}r$, $^{0.1}i$ and $^{0.1}z$) within a tight band of redshift at $z = 0.11$ for a number of cosmological models [16]. Their values for a Friedmann–Robertson–Walker (FRW) cosmological world model with assumed matter density $\Omega_0 = 0.3$, vacuum pressure $\Omega_\Lambda = 0$ and Hubble

constant $H_0 = 100$ h km s$^{-1}$ Mpc$^{-1}$ are presented graphically (Figure 3) and summarised in Table 2.

**Table 2.** SDSS Schechter function fits for $\Omega_0 = 0.3$ and $\Omega_\Lambda = 0$ cosmology at $z = 0.11$ [16].

| Band | $\phi^*$ ($10^{-2}$ h$^3$ Mpc$^{-3}$) | $M^* - 5\log_{10}$h | $\alpha$ |
|---|---|---|---|
| 0.1u | $3.26 \pm 0.40$ | $-17.89 \pm 0.04$ | $-0.94 \pm 0.09$ |
| 0.1g | $2.42 \pm 0.10$ | $-19.34 \pm 0.02$ | $-0.92 \pm 0.04$ |
| 0.1r | $1.69 \pm 0.06$ | $-20.37 \pm 0.02$ | $-1.03 \pm 0.03$ |
| 0.1i | $1.62 \pm 0.06$ | $-20.78 \pm 0.03$ | $-1.02 \pm 0.04$ |
| 0.1z | $1.47 \pm 0.05$ | $-21.12 \pm 0.02$ | $-1.07 \pm 0.03$ |

Blanton's results provide consistent and detailed parameters for the colour bands they selected, and, although they do not provide a subdivision by galaxy type, they have the advantage of uniformity in their selection by sampling a well-defined shell at a fixed galactic distance.

### 3.3. Number Counts from the SDSS

The extinction-corrected magnitudes and redshifts of a total of 1,424,733 galaxies were downloaded for all surveyed galaxies in the five principal filter colours of the surveys in the SDSS up to the survey cut-off magnitude of each band, with the number densities counted into 0.5-magnitude bands over a total sky area of 5954 square degrees (Figure 5). The counts show a systematic increase at all magnitudes from the ultraviolet towards the near-infrared.

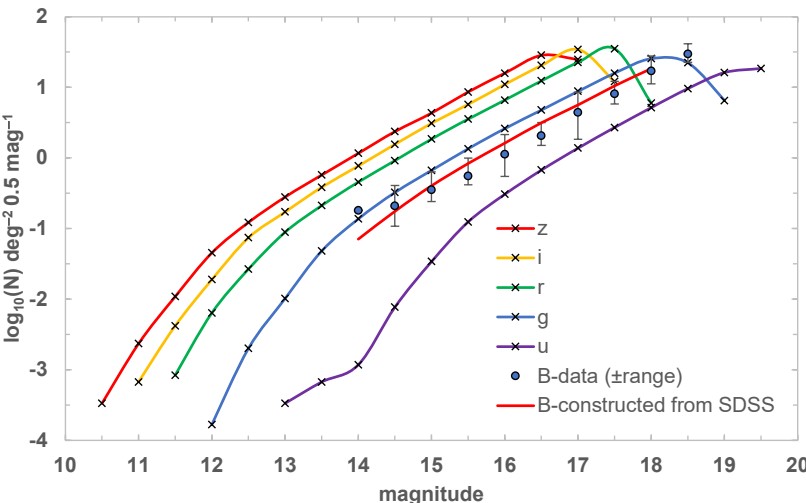

**Figure 5.** Galaxy number counts for the five SDSS observational bands (crosses). Overlain are the bright *B*-band counts (circles $\pm$ bin ranges) and the SDSS-derived *B* curve (solid red line).

Cross-referencing historical magnitude systems with the more recent SDSS filters is not a trivial problem [19], as the earlier B-filters were calibrated on stars based on the Johnson–Cousins UBVRI System, generally indicated by $B_J$. Empirical conversion to this system from the SDSS was performed from the $g'$ and $r'$ bands using Equation (10) [20], with appropriate de-reddening and k-corrections [21].

$$B_J = g + 0.155 + 0.152 \times (g - r) \tag{10}$$

This equivalence is shown in Figure 5 over the range 14–18 mag. (red line), where it overlays the $B_J$ observational number counts (circles) and falls within the range errors of these counts, confirming this to be an appropriate correction.

## 4. Observations in the Optical Bands

Extensive number count data for six principal observational bands, ultraviolet (*U*-), blue (*B*-), red (*R*-, *I*-), and infrared (*H*-, *K*-), with the Effective Wavelength Midpoint ($\lambda_{eff}$) for each standard filter at 365 nm, 445 nm, 658 nm, 806 nm, 1630 nm and 2190 nm respectively, have been compiled over many years from many sources [2,22–26] and are available at the Durham Number Count Survey site [1]. Over small redshifts, this volume is Euclidean, but as the counts probe deeper, the volume contained in each interval of redshift begins to shrink with increasing look-back time as observations come from earlier epochs of an expanding universe, while additional corrections must be made for luminosity evolution and mergers and the increasing density of galaxies within a reducing volume. A composite curve may then be constructed as in Figure 6 for the *K*-band.

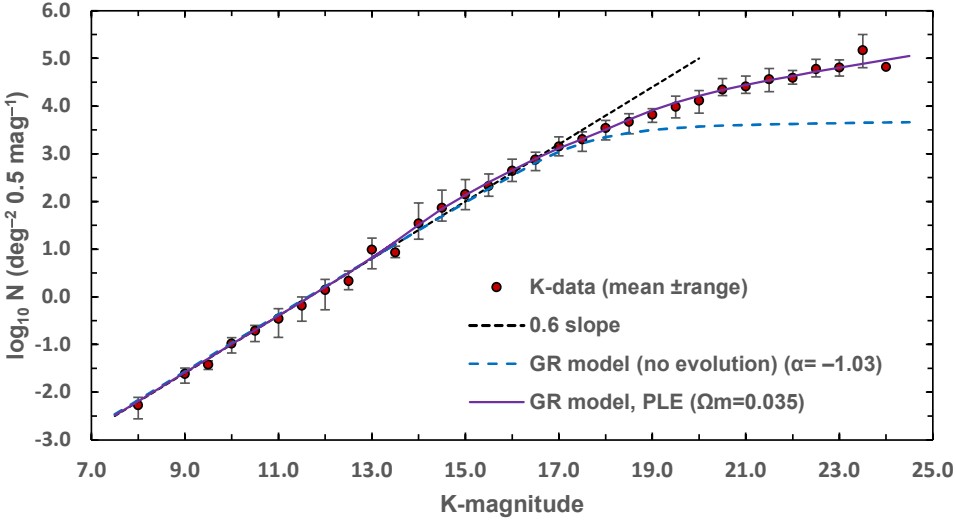

**Figure 6.** *K*-magnitude plots with no evolution (dashed line, $\alpha = -1.03$) and pure luminosity evolution (PLE) from $\alpha = -1.03$ to $\alpha = -1.70$ (solid line) in the selected GR model, both with $M^* = -23 + 5\log_{10} h$. The error bars reflect the maximum range of each bin; the absence of error bars implies the bin has a single member. The GR model fully overlays a $\Lambda$CDM model with $\Omega_m = 0.3$, $\Omega_\Lambda = 0.7$. The Euclidean 0.6 slope is also shown (dashed black line). Adapted from composite means of 42 *K*-band surveys from Durham number count archive [1].

The assumed initial values for $M^*$ and $\alpha$ for the model curves were fitted to the GR model using the SDSS data of Blanton et al. [16], assumed to be correct at $z = 0.1$, with $\Phi^*$ adjusted to normalise the curves to the observational data (Table 3). All curves were fitted by pure luminosity evolution (PLE) excepting the *B*-band curve, which required a starburst addition to create the sharp rise above $B_J \sim 22$.

**Table 3.** Schechter and PLE parameters for the six passbands of Figure 7, adapted from the SDSS parameters of Blanton et al. [16] ($H_0 = 100$ h km s$^{-1}$ Mpc$^{-1}$ with $h = 1$).

| Param | K | H | I | R | B | U |
|---|---|---|---|---|---|---|
| $\alpha$ | $-1.10$ | $-1.10$ | $-1.10$ | $-1.40$ | $-0.92$ | $-1.10$ |
| $M^*$ | $-23.20$ | $-23.50$ | $-22.00$ | $-22.10$ | $-20.73$ | $-22.00$ |
| $z_{max}$ | 1.00 | 1.20 | 2.00 | 2.00 | 2.30 | 1.10 |
| $\phi^*$ | 0.018 | 0.006 | 0.006 | 0.002 | $1.26 \times 10^{-4}$ | $1.96 \times 10^{-5}$ |
| $m_{norm}$ | 12.00 | 12.00 | 14.00 | 14.00 | 16.00 | 18.00 |
| $Z_{lim}$ | 2.0 | 3.0 | 3.2 | 3.0 | 4.0 | 5.0 |

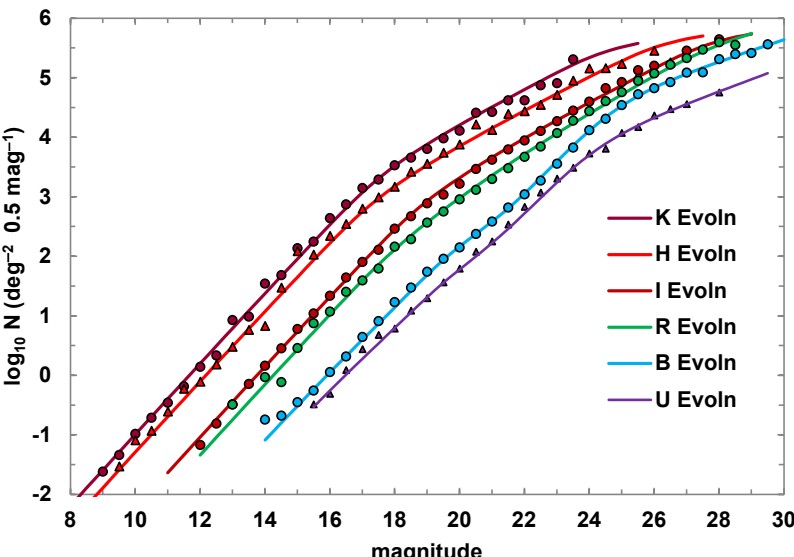

**Figure 7.** All six colour bands: GR model with PLE for *K*-, *H*-, *I*- and *R*-bands and starburst GR model for *B*- and *U*-bands. Composite means of all surveys (error bars omitted for clarity). (Data from Durham number count archive [1].)

### 4.1. K-Band Magnitude Count

A standard way to present all 42 published *K*-band series is to plot individual surveys overlapping along the curve. Figure 6 shows these data with all series binned into bins of 0.5 *mag* and plotting the mean of each bin. The error bars show the range of counts in each bin from the many surveys. The data for the *K*-magnitude-number counts demonstrate the theoretical asymmetric hyperbola of the model with an inflexion at $K_{mag} \simeq 17.25$. This binned representation gives a tight fit to the data points and is used to show the remaining bands (Figure 7).

There is still some controversy about the best parameters to use when building these models. De Propris et al. [27] suggested brighter $M^*$ but low alpha, such as $M^* = -24.63$ to $-24.48$ and $\alpha = -0.21$ to $-0.81$, while Huang et al. [28] determined $M_K^* = -23.57 \pm 0.08 + 5 \log h$, $\alpha = -1.33 \pm 0.09$ and $\Phi^* = 0.017 \pm 0.002 \, h^3 \, \text{Mpc}^{-3}$ for a universe with $\Omega_0 = 0.3$ and $\Omega_\Lambda = 0$, and Kochanek et al. [29] suggested $M^* = -23.39 - 5 \log h$, $\alpha = -1.09$ and $\Phi^* = 1.16 \times 10^{-2} \, h^3 \, \text{Mpc}^{-3}$.

The picture of a steepening $\alpha$ with redshift may be fundamental to reconciling the steep observed slopes of the number counts at faint magnitudes to the flatter LFs reported at the present epoch (or to $z = 0.11$ for the SDSS data). A pure luminosity evolution (PLE) model was therefore fitted to Figure 6, where the solid line shows the best fit for the GR model using a single Schechter model with modest PLE up to $z = 0.45$, with $\alpha$ increasing with redshift from $-1.03$ to $-1.70$ and $M^* = -23 + 5 \log_{10} h$. The curve was normalised at $K_{mag} = 10.5$ to the Frith 2MASS data points [30]. For comparison, the dashed curve shows the flatter $\alpha = -1.03$ with no evolution and a Euclidean line with a slope of 0.6. At these low redshifts integrating to $z = 0.45$, the Hubble GR model with $\Omega_m = 0.035$ overlaid the $\Lambda$CDM model with $\Omega_m = 0.3$, $\Omega_\Lambda = 0.7$ too closely to be distinguished.

### 4.2. The H-, I- and R-Bands

The *H*-band data (Figure 7), also plotted in bins of 0.5 mag, are from 14 surveys. The GR model with PLE shows a similar inflexion to the *K*-band, but with a flattening tail at faint magnitudes ($H_{mag} \sim 25$), with $M^* = -23.5 + 5 \log h$, $\alpha = -1.07$, and PLE beyond $z = 0.52$. The curve was again normalised to the Frith 2MASS (2006) points [30], which show a good Euclidean fit. The tail comes from cutting the integration limit at $z = 2.5$, but the plots can also be fitted to this tail using the values of Kochenek [29] with

$M^* = -23.9$, $\alpha = -1.03$, and integrating out to $z = 2$. Further data beyond $H_{mag} > 27$ may determine if the tail is real or whether the true counts continue to rise.

The *I*-band data (Figure 7) are from 26 surveys and show a gentle inflexion with a tailing off beyond $I_{mag} \sim 26$. The GR model mimics the data well, with $M^* = -21.00 + 5\log h$, $\alpha = -1.0$ and $\Phi^* = 0.0328\,\text{h}^3\,\text{Mpc}^{-3}$, as in the SDSS paper [16] and PLE. As with the *H*-band plots, a good fit to the tail is obtained by limiting the integration to $z = 2.0$.

The *R*-band data, taken from 31 surveys, show an inflexion at $R_{mag} \sim 24$, with a slight tailing off at faint magnitudes ($R_{mag} \sim 27$). Using the SDSS survey values $M^* = -20.37 + 5\log h$, $\alpha = -1.03$ and $\Phi^* = 0.0169\,\text{h}^3\,\text{Mpc}^{-3}$ with a PLE model to $z = 0.85$, the model mimics the data reasonably well with modest PLE (Figure 7).

The *U*-band data are widespread over much of their range, but these combined counts suggest mild starburst activity beyond $\sim$21mag, which is not evident in the raw count data, with a small inflexion beyond $U_{mag} > 22$. They include the recent data for the deep-*U* fields of Nonino et al. [31] to which the curve is normalised. These deeper-*U*-band data give significantly improved data in the redshift range $2 < z < 4$ and deeper colour-selected samples, such as the Lyman-break galaxies at $z \sim 3$. The GR model with $M^* = -22.0 + 5\log h$ and $\alpha = -1.10$ integrated out to $z - 4.0$ and with a starburst at $z \simeq 1.0$ lies within the error spread of the series. Figure 7 also includes the data and theoretical curve for the *B*-band, showing the sudden sharp rise in the counts beyond $B_J mag \sim 20$.

### 4.3. The B- and U-Bands

Ellis [32] described the *B*-band curves as an apparent excess of faint blue galaxy counts over the number expected on the basis of local galaxy properties. This has been referred to collectively as the faint blue galaxy problem [33] and is evident in Figure 8. This is constructed from 35 *B*-band surveys, again binned in half-magnitude bins, with the error bars showing the range of data in each bin and with the absence of error bars implying that the bin contains only a single data point. The swan-neck rise associated with the $B_J$ counts contrasts with the other curves and is emphasised in Figure 9, which plots the rate of change (slope) of the counts compared with the more regular *K*-band slope.

Metcalfe et al. presented number count data with a blue magnitude limit of $B_{mag} = 27.5$, showing that it was not possible to reconcile the slope and numbers of galaxies at $B > 25$ with the slope of the local faint galaxy luminosity function unless this was steeper in the past, or density evolution had taken place [23]. Metcalfe et al. [24] also performed a detailed analysis of more than 110,500 galaxies from the 2dF Galaxy Redshift Survey to generate an LF standardised to $z = 0$; over the interval $-16.5 > M_{B_J} - 5\log h > -22$, their LF is accurately described by a Schechter function with $M_{B_J} - 5\log h = -19.66 \pm 0.07$, $\alpha = -1.21 \pm 0.03$ and $\Phi^* = 1.61 \pm 0.08 \times 10^{-2}\,\text{h}^3\,\text{Mpc}^{-3}$, in broad agreement with earlier calculations by Yasuda et al. [34].

Modelling the number count curves in the *B*-band is confounded in part because of the ability to fit several different models to the same data by varying the parameters, and alternative models with an extensive void, PLE or a two-galaxy-type composite model can all be adjusted to a reasonable fit [5]. The best and easiest fit is a starburst evolution model shown for the GR model (blue line) and for a $\Lambda$CDM model (red line) in Figure 8. Parameters for the models were normalised to $B_J = 16$ with $M^* = -20.73 + 5\log_{10} h$ and $\alpha$ increasing from $-0.92$ to $-1.45$ integrated out to $z = 4.0$ for the GR model, and $M^* = -22.22 + 5\log_{10} h$ for the $\Lambda$CDM model. The starburst range was $0.6 < z < 1.2$, with the inflexion beginning at magnitude 20.5. This was in broad agreement with ultra-deep counts from the Herschel and Hubble Deep Field observations [25], although, at faint magnitudes, the $\Lambda$CDM model counts were too high, requiring an increase in $M^*$ to $-22.22 + 5\log_{10} h$ to tie it to the faint $B_J$ data.

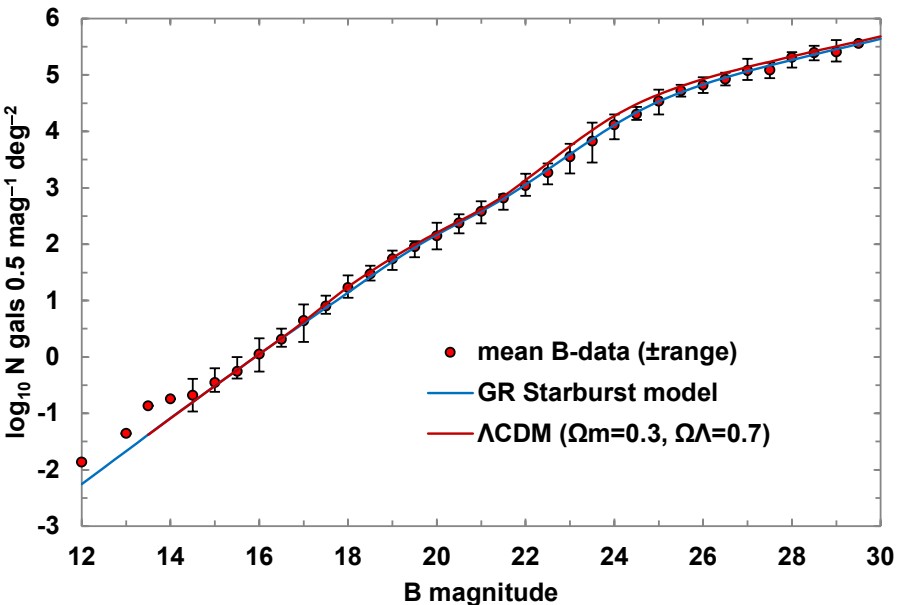

**Figure 8.** *B*-band magnitude counts binned $\pm 0.25$ m (composite means of 35 $B_J$-band surveys) overlain with the GR starburst model (blue line) with starburst evolution from $0.3 < z < 1.2$, and the $\Lambda$CDM model (red line). Bars indicate the range of observations in each bin. Adapted from Durham galaxy number count archive [1].

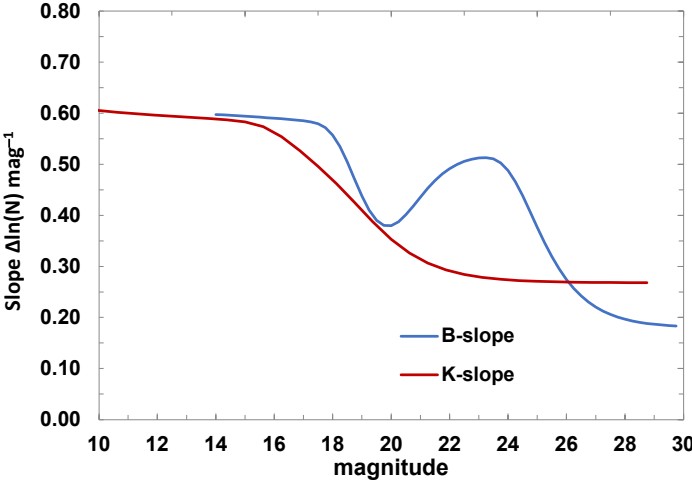

**Figure 9.** Change in slopes of *B*-band magnitude counts compared to *K*-mag counts.

The *U*-band data (Figure 7) were widespread over much of their range, but the combined counts suggest mild starburst activity, which is less evident in the raw count data, with a small inflexion beginning at $U_{mag} > 19.75$. They include the recent data for the deep-*U* fields of Nonino et al. [31] to which the curve is normalised. These deeper-*U*-band data give significantly improved data in the redshift range $2 < z < 4$ and deeper colour-selected samples such as the Lyman-break galaxies at $z \sim 3$. The GR model with $M^* = -21.9 + 5 \log h$ and $\alpha$ increasing from $-0.94$ to $-1.70$ and with a starburst in the range $0.6 < z < 0.97$ lies within the error spread of the series. Figure 7 also includes the data and theoretical curve for the *B*-band, showing the sudden sharp rise in the counts beyond $B_J mag > 20.5$.

## 5. Discussion

The observational data for GNCs have been accumulated over many years from a variety of sources and across different sky fields, causing much of the data to be widespread,

and GNCs in the six principal optical bands have traditionally been plotted as scatter plots to include each individual data point. In this paper, all counts were binned in 0.5 magnitude bins producing much cleaner data points for each band (Figures 6–8). All curves were well characterised by a Euclidean slope $N \propto 10^{0.6m}$ at bright magnitudes, with counts increasing with wavelength from the UV to deep IR at all magnitudes (Figure 7) and an inflexion point whose apparent magnitude also increased with wavelength.

Detailed GNCs were also plotted for the $u, g, r, i$ and $z$ colour bands of the SDSS and showed good correlation with counts in the optical $B$-band up to the SDSS cut-offs, using the conversion $B_J = g + 0.155 + 0.152 * (g - r)$. Data from the SDSS provide broad information on more than one million galaxies in five colour bands and provided detailed reference information for $\Phi^*$, $M^*$ and $\alpha$, the principal parameters of the Schechter function, at a well-defined redshift. The presence of voids can bias GNCs in near-surveys, confirmed by a deficiency in the count density/redshift between Northern and Southern galactic hemispheres of 26% out to a redshift $z \sim 0.15$ in the SDSS counts, as noted in earlier papers [18]. The steepening number counts at faint magnitudes reflect a much higher LF in the past. A simple merger model has been shown to reproduce this decline with a derived merger rate at $z = 1.5$, close to the observed value based on the increase in number densities, with most of these merging galaxies being of lower mass [35].

The need for faster rates of evolution in the $B_J$-band than the redder $R$- or $K$-bands may be explained by a model for evolution whereby some types of galaxies radiated larger amounts of flux in the rest $U$-band relative to the redder passbands [36]. Blanton et al. found that most galaxies in the SDSS with low redshift ($z < 0.05$) at the faint end of the LF are blue, and therefore, most dwarfs in the universe should be of the star-forming type with the same luminosity evolution as the star-forming galaxies (SFGs) [37,38]. Figure 10 illustrates the redshift at which each colour band begins to pick up galaxies at increasingly shorter wavelengths. It will be noted that the $B$-band moves into the UV by redshifts $z \simeq 0.2$, and the $B$-band inflexion, beginning at $B_{mag} \simeq 20.5$, corresponds to redshift $z \sim 0.5$, where this band is picking up galaxies in the deep UV but does not begin to pick up soft X-ray radiation until $z \simeq 10$.

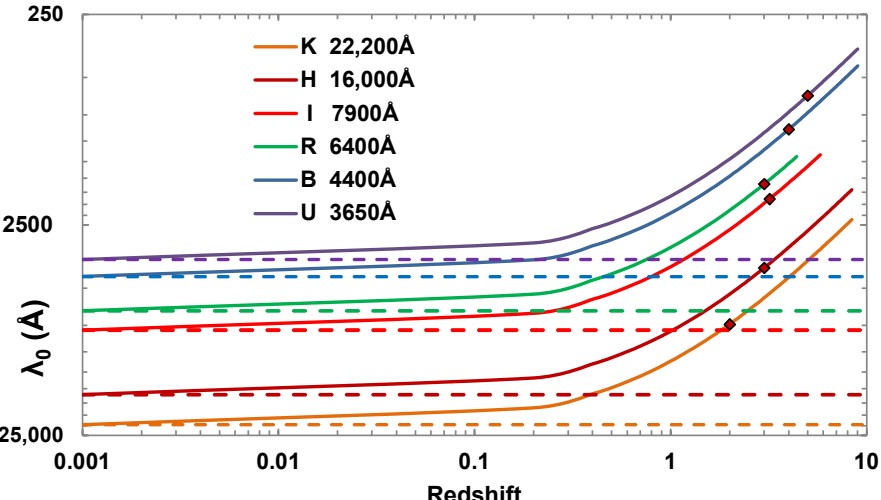

**Figure 10.** Log–log plot of the observed emission wavelenths vs. redshift for the six optical bands (solid lines) with reference lines for each band (dashed lines). Diamonds show the cut-off redshifts used to generate the number count plots for each band.

The giant ellipticals and spirals have been relatively stable since $z \sim 0.7$, whereas there has been rapid evolution of the irregular galaxy population with an increasing number of irregular and peculiar systems with increasing magnitude, with clear evolution of the blue population and little or no evolution of the red population [39]. The Hubble Space Telescope

Medium Deep Survey (HST-MDS) found that the universe is dwarf-rich at $z = 0.3 - 0.5$, consistent with a dwarf luminosity function with a steep faint-end slope, and a number of recent surveys have identified galaxies with stellar masses as high as $\sim 10^{11} M_\odot$ out to redshifts $z \simeq 10$ [4]. The GOODS-ALMA survey has detected a number of massive galaxies in the *H*-band with a median stellar mass of $M^* = 1.1 \times 10^{11} M_\odot$ and median near-infrared-based photometric redshifts of $z = 2.92$, with two of them ($z \sim 4.3$ and 4.8) suggesting redshifts $z > 4$. The ALMA survey also noted that $\geq 40\%$ of galaxies hosted an X-ray AGN, compared to $\sim 14\%$ for other galaxies of similar mass and redshift [40]. One further possible cause for this increase is an excess peak at $z \sim 0.69$ from gravitational lensing, which may inflate the true number count by as much as $\times 2.5$ out to $B \leq 27$, equating to an additional log count of $+0.4$ [41].

Starburst galaxies have an exceptionally high rate of star formation, up to 100 $M_\odot$ per year, compared with an average rate of star formation of only $\sim 3 M_\odot$ per year for a stable galaxy [42]. This is possibly driven by an epoch-dependent galaxy merger rate causing a combination of starbursts and weak AGN activity up to $z \sim 0.5$–1.0, and some support for such strong evolution comes from the microJansky radio source population, thought to originate from a mixture of starburst, post-starburst and elliptical galaxies [43]. Owen et al. have suggested that there is also extensive starburst activity at very high redshifts ($z \geq 7$), with extreme X-ray luminosity to explain the rise in observable counts detected in the *B*- and *U*-bands, where the excess UV and X-ray emissions are red-shifted into the blue and violet [44]. Sources making up the faint, sub-microJansky radio sky include SFGs, radio-quiet active galactic nuclei, low-radio-power ellipticals and dwarf galaxies. Kellermann et al. selected low-redshift SDSS quasi-stellar objects (QSOs) for an ultra-deep QSO survey and suggested that the QSOs primarily comprise two components: (1) a small fraction of quasi-stellar object (QSO) galaxies with radio-loud active galactic nuclei (AGNs) accreting into a central, supermassive black hole, characterised by their luminous, powerful radio emission, and (2) a large number of starburst-driven QSO host galaxies with a radio-quiet AGN population but an appreciable fraction of strong X-ray-emitting AGNs [45,46].

## 6. Conclusions

Although originally seen as a possible cosmological test, GNCs are now considered to be more sensitive to galactic evolution than to the cosmological model, and Ellis has stated that, "although the original motivation was an attempt to quantify the cosmological world model, [this is now] concerned with the study of faint galaxies as a way of probing their evolutionary history" [47]. The cosmological model selected is, therefore, of less importance than changing the rate of evolution to match the model chosen [47], and the general relativity (GR) model selected as the cosmological model for the count data includes Hubble expansion as an Einstein curvature because this has accurately modelled the luminosity distances derived from extensive supernovae (SNe 1a) data and the angular diameter distances derived from baryonic acoustic oscillation (BAO) data [6]. The Hubble GR model with $\Omega_m = 0.035$ could be fitted to all the GNC observational data with no intrinsic correction whereas, in comparison, the $\Lambda$CDM model with $\Omega_m = 0.3$, $\Omega_\Lambda = 0.7$ (Figure 8) fitted the sudden rise in *B* counts less well.

The Schechter models used for this paper followed Blanton [16] for the SDSS, using composite values for each colour band, and, although each galaxy type may have evolved with its own parameters and time scale, the curves described the observations sufficiently well that additional discrimination of the parameters for $M^*$ and $\alpha$ were not required. The four reddest wavelengths (*K*-, *I*-, *R*- and *H*-) required an evolutionary term in $\alpha$ to produce steeper faint slopes to fit the observations, with $\alpha$ evolving to reflect that there were more galaxies in past epochs, reducing through mergers to the present epoch, although $M^*$ and $\Phi^*$ will also have evolved, and evolution rates probably varied over the lifetime of these galaxies [25]. The two shorter wavelengths (*B*- and *U*-) each showed a sharp rise in their curves, particularly pronounced in the *B*-band, and these required a modest starburst

parameter with a rapid increase in $M^*$ ($\leq 1$ mag) to give a good fit over the whole range of observations.

The underlying physical cause for these rapidly evolving galaxies appears to be related to their star formation rates. Galaxies with the strongest [O II] emissions dominate the evolutionary trends, and their luminosity density has fallen by a large factor since $z \sim 0.5$ [48]. At the present epoch, such systems all lie at the faint end of the LF, but even at modest redshifts, they occupy a wide range of luminosities. Cohen [49] reported that *E* galaxies become more luminous by $z \simeq 1$, with the mean star formation rate (SFR) increasing strongly in the range $0 \leq z \leq 1$ by a factor of $\sim$10. The most luminous galaxies showed this trend very strongly, but Cohen reported only a modest evolution of $M^*$ with $z$, consistent with passive evolution.

Extending the SDSS count densities to higher redshifts and apparent magnitude limits showed a secondary peak in the counts at redshift $\sim$0.5, with many QSOs, AGNs, irregular dwarfs, starburst galaxies or giant elliptical galaxies. These are all active galaxies with high star-forming activity and correspondingly higher UV emission, leading to brighter absolute magnitudes ($M^*$) in the past [42,43]. Further support for a second peak at high redshifts has come from McDonald et al., who reported X-ray, optical and infrared observations of the galaxy cluster SPT-CLJ2344-4243 at redshift $z = 0.596$ that revealed an exceptionally luminous galaxy cluster that appears to be experiencing a massive starburst, with a formation rate of $\sim$740 solar masses per year [50]. Atek et al. have probed the UV LF to $z \sim 7$, computing each field individually and comparing their results to the compilation of the *HST* legacy fields, the LF fields and the UDF12 field [51]. One further important possible cause for this increase is an excess peak at $z \sim 0.69$ from gravitational lensing, which may inflate the true number count by as much as $\times 2.5$ out to $B_{lim} \leq 27$, equating to an additional log count of $+0.4$ [41].

The 1–5 μm range from the JWST observations has allowed a search for intrinsically red galaxies beyond the $\lambda_{rest} = 1216$ Å Lyman break and the $\lambda_{rest} \sim 3600$ Å Balmer break in the first $\simeq 750$ Myrs after the Big Bang, and six possible galaxies at $7.4 \leq z \leq 9.1$ with $M > 10^{10} - 10^{11} M_\odot$ have now been found, suggesting that the stellar mass density in massive galaxies may be much greater than predicted by the rest-frame UV-selected samples [52]. Finkelstein et al. [53] have identified a sample of 26 galaxies in the first 500 Myr of galaxy evolution at $z \sim 9$–16 from the Cosmic Evolution Early Release Science (CEERS) JWST survey. These objects were compact with a median half-light radius of $\sim$0.5 kpc and a surprisingly high density, and a high-redshift galaxy at $z \sim 11.09$ that is extremely luminous in the UV with a mass of $\sim 10^9 M_\odot$ has also been observed [54]. An estimate of the $z \sim 11$ rest-frame UV luminosity function found that the number density of galaxies (arcmin$^{-2}$) showed little evolution from $z \sim 9$ to $z \sim 11$, and Mcgaugh has noted that early, large galaxies at $z \geq 10$ may contribute to these very bright number counts [55].

The new observational challenge is simultaneously to cover the low-mass and high-redshift ranges of the galaxy population, especially at $1 < z < 4$ where galaxies are most actively forming stars. As most spectral features move into the optical rest-frame at these redshifts, deep near-infrared (NIR)- and infrared (IR)-selected data will be essential for accurate photometric redshifts and stellar masses, to probe differing galaxy environments, to accurately trace the large-scale structure and to minimise the effect of cosmic inhomogeneities [56]. For the future, kinetic Sunyaev Zel'dovich (kSZ) tomography hopes to provide useful constraints on some bias parameters in galaxy number counts and to enable a clearer distinction between standard ΛCDM cosmology and relativistic Hubble expansion cosmology [57]. New cosmological probes based on re-scattered CMB photons will enable huge galaxy surveys to map vast volumes of the observable universe to the extent that general relativistic corrections to the distribution of galaxies must be taken into account, and the JWST, ranging from visible through the mid-infrared, covering 0.6–28 μm, may reveal the assembly of galaxies going back to the first light in the early universe.

**Funding:** This research received no external funding.

**Data Availability Statement:** Optical data available at the Extragalactic Astronomy and Cosmology Research Group of Durham University [1]. Sloan Digital Sky Survey data available at sdss4.org/.

**Acknowledgments:** I would like to thank the anonymous reviewers for their constructive criticism; Tom Shanks and Nigel Metcalfe for their early support; the Extragalactic Research Group of Durham University for collating and publishing their many observations; the SDSS team for making their data fully accessible; and the IoA at Cambridge for their facilities.

**Conflicts of Interest:** The author declares no conflict of interest.

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
