# Peer review of "Faint Galaxy Number Counts in the Durham and SDSS Catalogues"

_galaxies, doi:10.3390/galaxies11030065_

Round 1

Reviewer 1 Report

The manuscript presents an analysis of galaxy number counts in several photometric bands, from u up to K. Despite the title, it does not focus only on the data from the Sloan Digital Sky Survey (SDSS), but instead a significant fraction of the paper is based on the Durham Extragalactic Astronomy and Cosmology catalogue. The author compares the number counts from these datasets to predictions of a cosmological model together with modelling luminosity evolution, via the luminosity function.

This could potentially be valuable work, however at present the paper requires quite a bit of work to be publishable. Below I list the main issues I see in this manuscript, although there are also several more minor ones which I do not discuss.

* The two datasets used – ‘Durham’ and SDSS – do not seem to be related to each other. Also in the paper, it is not clarified how the analyses based on them are linked. Section 6 focuses on the Durham data, and then in Section 7 SDSS is discussed separately. It would be good to provide a better connection between these studies.

* The Durham data are more than a decade old compilation. Is this the most recent knowledge in this matter? The most recent paper quoted in this context is from 2006, while some of the works contributing to the Durham-based number counts are from1991. I believe there has been quite a bit of progress in surveying the deep Universe in the various wavelengths since then. A good example is the research of the COSMOS field.

* The author does not seem to account for various selection effects related to the spectroscopic SDSS data used. The Sloan spectroscopic galaxy sample is complete (flux-limited, modulo fibre collisions) only to r<17.77 (SDSS Main Sample). Beyond this limit, galaxies for SDSS spectroscopy were selected based on their colours (LRG, CMASS, BOSS, eBOSS, ...) and do not provide a uniform selection, which is an important aspect for a number count analysis. As no compensation for related incompleteness is offered in the manuscript, the discussion of SDSS number counts beyond r>17.77 is of little value. In particular, the shape of the SDSS redshift distribution in Figure 14 illustrates this issue very well. The first peak comes from the SDSS Main Sample, while the 2nd and 3rd peaks reflect the SDSS selection function. This has very little to do with the general galaxy distribution (and hence number counts) in the Universe. Because of that, most of the discussion in Sec.7.1 is unfounded and plots such as Fig.15 tell us nothing about the underlying number counts of galaxies beyond bJ~18 (which corresponds to r~17.77 I suspect).

* To compute cosmological distances and volumes, a cosmological model with zero cosmological constant is used. While this is of less importance in the local volume (z<0.1), it becomes crucial at higher redshifts. Some discussion is needed on how the calculations, results and conclusion would differ if instead the LCDM model was adopted. Let me here note that there is overwhelming evidence from a number of probes that Omega_Lambda>0 in our Universe.

* What is novel in Sec.3, how does this differ from the standard textbook knowledge?

* The reference band used for a large part of the number count study is the photographic bJ. First, what is the reason for adopting this obsolete band rather than taking something more modern, such as g or r? Second, how are the various other passbands converted to bJ, in particular the SDSS u,i,z ones? Some equations and references would be useful. At the moment only the equation to convert g,r to bJ is provided.

* The abstract says “The comparative densities confirmed an extensive void in the Southern sky

with a deficit of 26% out to a redshift z ∼ 0.15.” This void is later mentioned in the text, but I do not find any discussion on how it was confirmed in the present analysis, not to mention further details.

* There is inconsistency in the terminology. Normally, “number counts” would refer to the number of galaxies per unit observed magnitude (flux) per unit volume. This is indeed shown in Figs. 6-13 and 15, and discussed in the text. However, the same term “number counts” is applied to Figs. 2 & 14, which have the redshift z on the x-axis. I would call these plots “redshift distributions”, n(z) (or better, dN/dz, as these are differential). They are related to the number counts in an indirect way, via the selection function of a given survey plus the underlying redshift-dependent luminosity function.

A few notes on that occasion:

- Fig. 2 has no normalisation to the sky area – no wonder that the counts for S are so much smaller than for N, the former has much smaller sky coverage then the latter.

- Fig. 3 seems to be Fig.2 but now normalised by the sky area and presented as connected points rather than bars. These are now called “galaxy density” while still being redshift distributions. Then again Fig.14 calls such a redshift histogram “number counts”. Please adopt consistent terminology.

- Why such specific x-tics in Figs.2 & 14, starting from 0.01? Also, the interval between the numbers on x-axis in Fig.2 is not constant, in most cases it is 0.04, but sometimes it is 0.03.

* Please clarify if uppercase passbands are the same as lower case. As far as I know, for instance (photographic) R is not the same as (Sloan) r, etc.

* Magnitudes need to be anchored to some zero-point. Which convention was used, Vega or AB, and if conversions were applied from one system to the other, what equations were used?

* Some further comments on the SDSS analysis. What was the reason to cross-match SDSS galaxies with Simbad? SDSS provides a large database on galaxy properties, in particular in its spectroscopic samples galaxies are classified based on their spectra. In addition, was the sample used from SDSS composed of both galaxies and quasars? These classes are also clearly marked in SDSS. Last but not least, is it safe to derive general conclusions from a sample of just 100 galaxies?

* What is the connection between what is discussed in the three final paragraphs of Sec.7 and the rest of that section and of the paper? Especially the discussion on radio galaxies and QSOs does not seem to be related to the rest of the text.

* The title is misleading as SDSS is only one of the datasets used for the number counts.

Reviewer 2 Report

The paper describes the galaxy number counts from the Sloan Digital Sky Survey. It contains a detailed analysis in different photometric bands, highlighting the limitations for the interpretation of galaxy counts. The scientific questions are important and addressed in details; the manuscript is well written. In my opinion, the Introduction and the presentation of the results need some improvements before it can be published.

Introduction. In my opinion the introduction lacks a clear description of the aims of this paper and a summary of previous works from the literature studying this topic. Stating the hypothesis in the Introduction and how the paper stands out from the previous literature will help the reader to better understand the paper's goals. Many references are reported in the section "Evolutionary effects on the LF from the SDSS" and in the Discussion, however some of them are more appropriate for the Introduction and they will help the reader to assess the context of the paper.

Presentation of the results. Figures 7-12 can be grouped as different panels (Panel A,B,C..) of the same Figure. Having the panels on the same x- and y-axes and close to each other will help the eye of the reader to catch qualitatively differences. A summary table listing the various parameters related to the difference bands will also be useful and it will be an important reference for the reader. The table should contain the values for M*, alpha, Phi*, z_max, the used references and a short description of the trend/additional models.

Minor details:

Figure 1: add number of galaxies for each redshift range.

Line 56: explicitly name Phi* will help the reader with the context of the paper.

Line 56-57: description of Figure 2 can be summarized at lines 49-50.

Line 74: please add a reference related to BO.

Line 75: please add a reference related to SNe 1a.

Line 91: it would help the reader to state the values of z_max used in the paper.

Line 104: please specify "galaxy clusters".

Line 142-143: this is a repetition of the sentence at 125-126.

Line 177: I think it is the blue line?

Line 194: please add a reference for Frith 2MASS (2006) data points.

Line 231: please explicitly write what q0 is, as it will help the reader with the context.

Line 268-270: please quantify the used maximum redshifts as they might be an helpful reference for the reader.

Figure 13: it would be useful to plot the predicted Euclidian slope of 0.6 as reference.
